https://doi.org/10.1038/s41467-019-09740-7　　**OPEN**

# Microfluidic multipoles theory and applications

Pierre-Alexandre Goyette[1], Étienne Boulais[2], Frédéric Normandeau[3], Gabriel Laberge[2], David Juncker ● [3] &
Thomas Gervais[1,2,4]

Microfluidic multipoles (MFMs) have been realized experimentally and hold promise for "open-space" biological and chemical surface processing. Whereas convective flow can readily be predicted using hydraulic-electrical analogies, the design of advanced microfluidic multipole is constrained by the lack of simple, accurate models to predict mass transport within them. In this work, we introduce the complete solutions to mass transport in multi-polar microfluidics based on the iterative conformal mapping of 2D advection-diffusion around a simple edge into dipoles and multipolar geometries, revealing a rich landscape of transport modes. The models are validated experimentally with a library of 3D printed devices and found in excellent agreement. Following a theory-guided design approach, we further ideate and fabricate two classes of spatiotemporally reconfigurable multipolar devices that are used for processing surfaces with time-varying reagent streams, and to realize a multistep automated immunoassay. Overall, the results set the foundations for exploring, developing, and applying open-space microfluidic multipoles.

[1] Institut de Génie Biomédical, École Polytechnique de Montréal, Montréal, QC H3T 1J4, Canada. [2] Department of Engineering Physics, École Polytechnique de Montréal, Montréal, QC H3T 1J4, Canada. [3] Biomedical Engineering Department and Genome Quebec Innovation Centre, McGill University, Montreal, QC H3A 0G1, Canada. [4] Institut du Cancer de Montréal, Centre de Recherche du Centre Hospitalier de l'Université de Montréal, Montréal, QC H2X 0C1, Canada. These authors contributed equally: Pierre-Alexandre Goyette, Étienne Boulais. Correspondence and requests for materials should be addressed to T.G. (email: thomas.gervais@polymtl.ca)

O ver the past decade, "open-space" systems that provide locally addressable fluid streams have emerged and broadened the definition of microfluidics to include channel-free fluidic processors[1,2]. Contrary to traditional channel-based microfluidic systems, they operate from above a surface, are contact free, and can be used for local processing of large surfaces such as petri dishes and culture flasks with high resolution, which previously could only be accomplished using static, flow-less methods such as inkjet and pin-spotting. Pioneering technologies such as the microfluidic pipette[3], aqueous two-phase reagent delivery[4], and droplet-based reagent delivery and sensing (a.k.a. the chemistrode[5]) all localize fluid flow without closed channels. Arguably the most established, most versatile open-space microfluidics technology to date is the microfluidic probe (MFP)[6], a device originally containing two flow apertures, one of which is withdrawing a fluid. The apertures of a MFP are located within a blunt tip positioned parallel, close to a surface, effectively forming a Hele-Shaw cell[7]. By modulating the flow ratio between injection and aspiration apertures, while keeping a net positive aspiration under the device, confined streams of reagents under the MFP can be scanned over the surface to form patterns with high spatial resolution, low shear stress, and low reagent consumption. MFPs, however, face one main drawback: their scanning speed is limited by the reaction kinetics between the delivered reagent and the surface. Reaction times in the life sciences being generally in the minute to hour time scales, surface patterning with a MFP becomes slow and impractical in many instances due to the inherently serial nature of the scanning process.

To increase MFP versatility, a growing number of designs incorporating multiple flow apertures have been reported. They are all part of a more general class of flow patterns which we term microfluidic multipoles (MFMs). Under this nomenclature, the original two-aperture MFP design can be construed as a simple microfluidic dipole[8]. Another well-studied MFM is the microfluidic quadrupole, which enables the simultaneous confinement of two different reagents[9]. However, the concept of open MFM is generalizable to an arbitrary number of injection and aspiration apertures, which may be configured to generate a variety of flow and diffusion patterns. We enumerated a total of 11 previously published different MFM configurations that are irreducible, i.e. they generate reagent profiles that cannot be achieved with one of the other systems. They were used for various processes, including surface functionalization[6,10], local cell lysis and DNA analysis[11,12], sharp gradient generation[13], tissue staining with immunohistochemical markers[14], and "Stokes trapping" of microparticles in large chambers acting as Hele-Shaw cells[15] (see Supplementary Table 1). While these represent a growing diversity, innovation in open-space microfluidics has so far mainly been driven by trial and error, which can be in part ascribed to the lack of a complete formalism to describe mass transport in 2D MFMs.

Several attempts have been made to model the flow and diffusion under open-space microfluidic devices. Full 3D finite element simulations have been used extensively[11,12,16]. However, they provide minimal insight on the relationship between design and operation variables and are too slow and resource-intensive to be used in a closed-loop, real-time experimental setup. From an analytical standpoint, the flow streamlines generated by point source openings located within a Hele-Shaw cell are rigorously analogous to the electric field lines around a distribution of point charges in 2D space[9]. Although seldom used in the context of microfluidics, this analogy effectively generalizes the oft-used hydraulic-electrical analogy to model the pressure-flow rate relationships in networks of quasi-1D microchannels using Kirchhoff's laws[17]. However, contrary to the case of simple parallel streams inside a microchannel, taking the diffusion of a scalar (concentration, temperature) into account in a 2D flow field remains a challenge due to their typical complexity. As a result, despite over a decade of efforts, a complete analytical expression for 2D advection-diffusion profiles in MFMs is still missing, even for the dipole, the simplest open-space microfluidic unit and canonical embodiment of the MFP. Moreover, the few approximations published to this day are only valid for very localized areas of space and fail to account for the full diffusion footprint of the device.

On the experimental level, several MFM fabrication methods were presented over the years, but they all required microfabrication of silicon or glass components[6,18,19] which are costly and slow to yield prototypes. In recent years, 3D printing has emerged as a disruptive technology for microfluidic device fabrication[20–22]. 3D printing is fast and yields ready-to-use devices that require no alignment, bonding or molding steps. Moreover, single block printing of microfluidic devices affords full design flexibility in three dimensions that can simply not be realized using microfabrication processes dependent on iterative photolithography and microstructuring cycles. We recently presented a method for 3D printing of MFMs[23], but a proof of concept for the fabrication of more general open-space microfluidic devices is still missing.

Here, we first introduce an analytical framework to study the general problem of advective-diffusive transport in MFMs that is experimentally-validated using 3D printed MFM devices. The model we propose exploits mathematical advances in the conformal mapping of non-harmonic functions[24] to find transport solutions to infinite families of MFM with arbitrary number of apertures. In a second step, we employ our formalism and experimental platform and combine it with flow modulation to introduce spatiotemporally reconfigurable MFM devices which exploit the various symmetries in multipolar flow patterns. Whereas MFPs were scanned on surfaces, MFMs use the dynamic control of independent confinement zones to address multiple surface regions in parallel, effectively forming a 2D reconfigurable reagent display. Finally, the potential of MFMs for long-lasting multistep experiments is demonstrated by performing a fully automated, three-step immunofluorescence assay over an open surface, generating a complete binding curve in a single experiment.

## Results

**Model of advection-diffusion for an asymmetric dipole.** In this section, we lay out the basic transport theory for 2D MFMs as illustrated in Fig. 1. We use complex potential representation to provide a complete analytical model for both advection and diffusion profiles under an asymmetric flow dipole (sometimes called "doublet" in fluid mechanics) in a Hele–Shaw cell such as those formed by a dipole MFPs (Fig. 1a). In the subsequent section, we show how this model can be extended to obtain exact flow profiles for an infinite variety of MFM devices (Fig. 1b–d).

We define an asymmetric dipole of finite dimensions as two point-source apertures of arbitrary flow rates, one aspirating and one injecting, separated by a distance $L$. We assume an infinite flow domain, which in practice is realized by ensuring that the surface of the injection device is multiple times larger than the maximum inter-aperture distance. An adimensional variable system with $x = X/L$, $v = 2\pi GLV/Q_0$, $c = C/C_0$ is used. $X$ and $V$ are, respectively, the position and velocity vectors, $L$ is the interaperture distance in the dipole, $G$ is the height of the gap forming the Hele–Shaw cell, $Q_0$ is the injection rate of the injection aperture, and $C_0$ is the injected reagent concentration. We model the apertures as point sources and thus neglect their

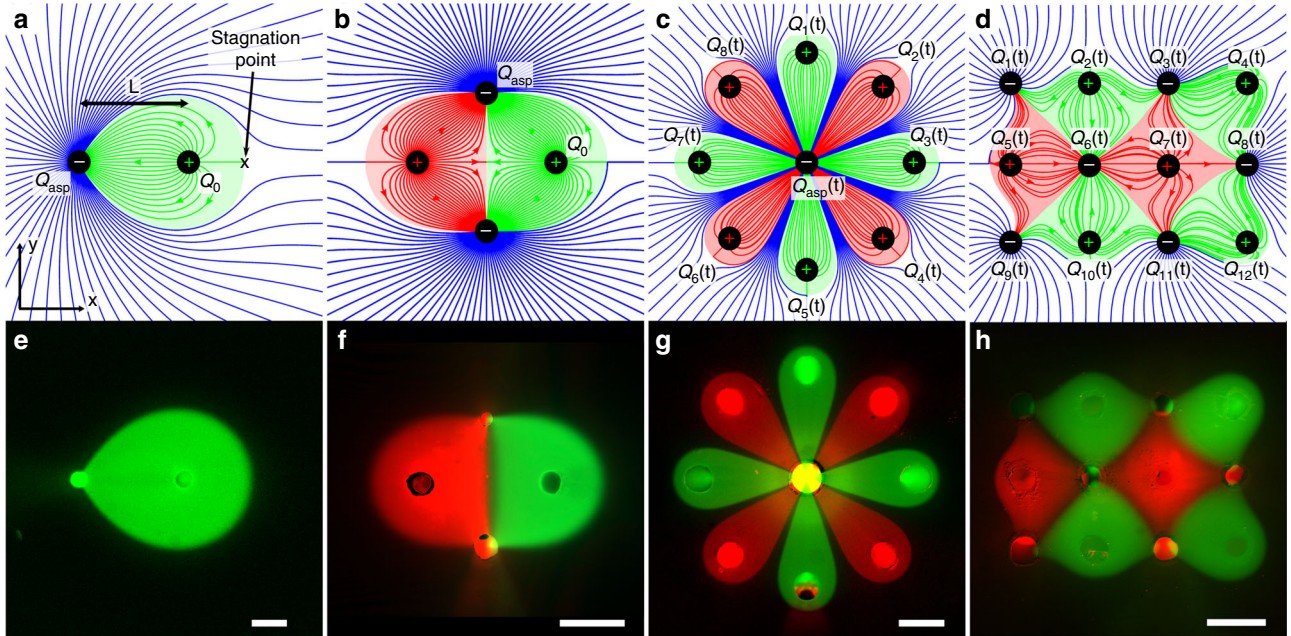

**Fig. 1** From dipoles and quadrupoles to multipoles. Theoretical streamlines (**a–d**) and fluorescence micrograph (**e–h**). Positive and negative sign, respectively, represent injection and aspiration apertures. To facilitate comparison between theory and experiment, green and red background were superposed on flow fields to highlight the expected areas of confinement not captured by streamlines. **a**, **e** Microfluidic dipole. **b**, **f** Microfluidic quadrupole. **c**, **g** MFM with rotational symmetry. **d**, **h** 12-aperture MFM with translational symmetry. Scale bars represent 500 μm

finite radii. The effects of this approximation have already been well-studied[25] and shown to be negligible in most practical applications. Creeping flow (Reynolds number « 1) is assumed throughout the analysis.

We use complex flow representation to describe vectors in the 2D plane $\mathbb{R}^2$ as complex numbers $z = x + iy$. This notation is already well-used in the fields of groundwater flow[26], viscous fingering[27], or in the design of airfoils and hulls[28]. A study of water permeation in bulk PDMS[29] constitutes its only application in microfluidics to the best of our knowledge. Under complex representation, a multi-aperture flow can be conveniently described by the complex potential[30]

$$\Phi = \sum_i q_i \log(z - z_i), \qquad (1)$$

where each point-like aperture is located at position $z_i$ and has flow rate $q_i$. One useful feature of the complex potential $\Phi = \phi + i\psi$ is that its real part describes the pressure field while the imaginary part represents the streamlines of the flow[17]. Furthermore, the potential (Eq. 1) can be differentiated to obtain the complex conjugate of the velocity flow field $\bar{u} = \frac{d\Phi}{dz} = u_x(x, y) - iu_y(x, y)$. This format enables the use of conformal mapping, which via a complex variable transformation of the form $\omega = f(z)$ warps the solution domain of specific 2D differential equations in a simple geometry to generate exact solutions for more complex geometries. Conformal mapping stems from the conformal invariance of the Laplace equation[31] and is used extensively to study purely advective multipolar flows in porous media[26]. Once the complex potential for a given problem is known, the diffusive transport of a diluted species within this field can also be obtained by solving the advection-diffusion equation under 2D potential flow. In cases where total aspiration is superior to injection, there exists a steady-state where advection balances diffusion, allowing us to use the steady advection-diffusion equation

$$\nabla^2 c - \mathrm{Pe} \, \nabla\phi \cdot \nabla c = 0, \qquad (2)$$

where $\mathrm{Pe} = Q_0/2\pi GD$ represents the ratio of diffusive to convective time scales. The algebraic term $\nabla\phi \cdot \nabla c$ constitutes a challenge as it quickly renders the equation intractable even for relatively simple flow patterns. To address this issue, we turn again to conformal mapping. It is known that the advection-diffusion equation for potential flows is, like Laplace's equation, one of a handful of conformally invariant PDEs[24]. Hence, the same conformal transformations could be applied to transforming advection diffusion problems into a streamline coordinates as originally proposed by Boussinesq[32]. Indeed, under this type of hodograph transform, the flow becomes straight and advection naturally becomes decoupled from diffusion, thus leading to a simplified transport equation:

$$\frac{\partial^2 c}{\partial \phi^2} + \frac{\partial^2 c}{\partial \psi^2} = \mathrm{Pe} \frac{\partial c}{\partial \phi} \qquad (3)$$

Using the streamline coordinates described above, the advection-diffusion profile under a dipole flow (Fig. 2b) can be represented easily in dimensionless units, with an injection aperture ($c = 1$) located at the origin, an aspiration aperture at z $= -1$ and a fixed concentration $c = 0$ for $|z| \to \infty$. The ratio of aspiration to injection flow rates is given by the parameter $\alpha = \frac{q_{\mathrm{asp}}}{q_{\mathrm{inj}}} > 1$. The flow pattern in such a dipole has a stagnation point located at[8]

$$z_{\mathrm{stag}} = \frac{1}{\alpha - 1} \qquad (4)$$

Upon inspection, the problem can be transformed to streamline coordinates (Fig. 2a) using the function

$$\Phi(z) = \log(z) - \alpha \log(z + 1). \qquad (5)$$

In the streamline domain $\Phi$, the problem is equivalent to a channel geometry with flows of concentration $c = 1$ and $c = 0$ separated by a no-flux boundary condition on the origin. At the stagnation point, the no-flux condition is dropped, and the flows are free to mix (Fig. 2a) (see Supplementary Table 2 for more

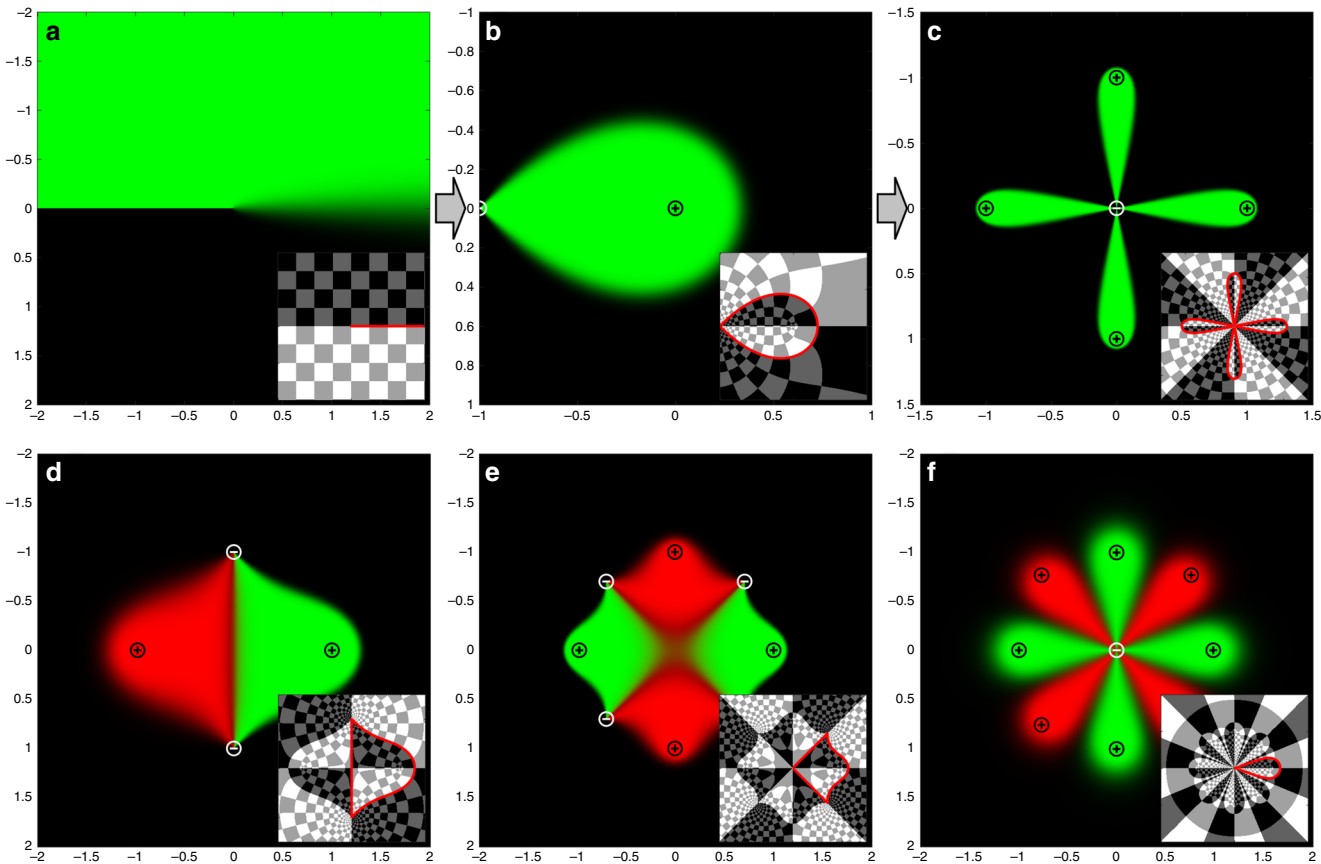

**Fig. 2** Theoretical model. Solutions for a leading edge in a no-slip plane flow (Pe = 100) is first obtained (**a**) and then transformed via the complex potential to obtain the dipole concentration profile (**b**). This solution can then be further transformed to obtain symmetrical configurations such as the "flower multipole" (**c**). Similar steps can be taken to obtain solutions for a variety of problems. Pictured here are (**d**) the microfluidic quadrupole (**e**) the "poppy" alternating multipole and (**f**) a multicolor "flower multipole" with different injected reagents. Black and white maps in inset represent the mapping of the upper complex plane and lower complex plane of solution (**a**), respectively. Checkerboard insets demonstrate how the transform used affects a regular grid. The red line corresponds to the line of concentration 1/2 and separates the "interior" and "exterior" domains

details on the streamline problem). The separating streamline going from the stagnation point to the aspiration aperture corresponds to the semi-infinite segment of the horizontal axis where the fluids can mix. If the Péclet number is high enough (higher than about 10, which is always realized in microfluidics applications), this segment can be taken to have concentration $c = 1/2$ and the walls of the channel geometry can be safely ignored (Supplementary Note 1). The problem can thus be decomposed in two problems of advection-diffusion around semi-infinite obstacles of fixed concentration. The problem of advection-diffusion around such a semi-infinite obstacle has been extensively studied in theoretical fluid mechanics, notably in the theory of dendrite solidification[33], and in the study of out of plane flow in Burgers vortex sheets[24,34]. It yields the solution

$$c(\Phi) = 1/2 \left( 1 \pm \mathrm{erf}\left( \mathrm{Im}\sqrt{\mathrm{Pe}\left(\Phi - \Phi_{\mathrm{stag}}\right)} \right) \right), \qquad (6)$$

where $\Phi_{\mathrm{stag}}$ is the image of the stagnation point and $\mathrm{erf}(x)$ is the error function[35]. The sign of $\pm$ is determined by whether we have an incoming flow of concentration $c = 0$ or $c = 1$. However, neither of these concentration profiles represent the full dipole footprint when transformed. This can be seen physically in the flow dipole, in which there is both incoming fluid at concentration 0 (aspirated from the system's surroundings), and incoming fluid at concentration 1 (injected by the aperture). To solve this issue, we separate the problem into an "interior" and an "exterior" domain at the streamline of concentration $c = 1/2$ (see

checkerboard insets in Fig. 2). There remains a discontinuity in our solution due to the branch cut of the logarithm functions in Eq. (1), but the solution can be made continuous by placing the singularities on the real axis and using it as an axis of symmetry. The final step is then to obtain the entire solution as a piecewise function assembling the "interior" and "exterior" solutions, given by transforming Eq. (6) back to the dipole flow domain $Z$. The interior and exterior domains can be defined either by checking the sign of $\Phi$ in the streamline domain or by using the expression for the separating line in the $Z$ domain in polar coordinates (see Supplementary Note 2).

This gives us the complete, exact expression for the concentration profile in the asymmetric dipole of finite dimensions

$$c(z) = 1/2 \left( 1 \pm \mathrm{erf}\left( \mathrm{Im}\left\{ \sqrt{\mathrm{Pe}\left( \log(z) - \alpha \log(z+1) - \Phi_{\mathrm{stag}} \right)} \right\} \right) \right). \qquad (7)$$

This expression is compact under complex representation and valid for high Péclet numbers.

**Generalization from microfluidic dipoles to multipoles.** Once a solution is known for one particular multipole flow profile, the full power of conformal transforms can be exploited. By using simple functions expressing symmetry operations, such as inversions or power transforms with a suitably placed origin, we can obtain concentration profiles for an infinite family of

multipoles. In each case, the transport problem is first solved in the streamline domain (Fig. 2a), then transformed to obtain the flow profile for an asymmetric dipole (Fig. 2b). The dipole solution can then be transformed again to obtain the desired flow patterns. For example, a power law conformal transform generates a polygonal structure with a rotation symmetry whose number of sides is dictated by the exponent of the power transform (as in Fig. 2c). These devices will have injection and aspiration apertures located at new positions, determined by the transform of the initial aperture locations, and can then be fabricated and operated to obtain the predicted patterns. This method gives us a comprehensive toolbox to not only model and explain phenomena in known open-space MFMs in terms of simpler ones, but also to explore new configurations that have not been investigated yet.

While many geometries can be obtained by directly transforming the concentration profile from the microfluidic dipole, an arbitrary placement of injection and aspiration apertures will in general not be reducible to a semi-infinite absorbing leading edge (Fig. 2a). In the more general case, the streamline coordinate problem will exhibit any number of finite and semi-infinite absorbing segments. By solving these streamline coordinate problems, we obtained new families of multipolar devices, such as the two-reagent microfluidic quadrupole (Fig. 2d), which can itself be transformed to obtain new symmetrical patterns (Fig. 2e, f and Supplementary Note 3 for more details).

Table 1 summarizes some common transforms that can be used to obtain the concentration profile for many new devices from the dipole solution without solving any differential equation. A more detailed version, including many more transformation groups is presented in the SI. These configurations include known devices such as the previously published microfluidic quadrupole, but also several geometries that have not yet been investigated such as "flower" multipoles (Fig. 2c, f), alternating multipoles (Fig. 2e), polygonal multipoles and impinging flow multipoles (Table S3),

**Practical implementation of fixed microfluidic multipoles.** To validate our theoretical model, multipolar microfluidic devices were fabricated using a previously published single step 3D rapid prototyping process[23], which offers several advantages over conventional silicon-based machining: it is fast and simple, and truly 3D structures can be produced, which allows the fabrication of compact MFM with high density of fluidic ports that are connected via complex fluidic routing architectures to arbitrarily positioned apertures. MFM devices can be operated in scanning probe mode, as MFPs. They can, in addition, be operated while remaining stationary above the surface (Fig. 3a). Under this "fixed" mode, an accurate gap height between the MFM and the substrate is ensured via integrated 3D-printed spacers, instead of expensive mobile parts (Fig. 3a and Supplementary Fig. 4). MFM devices are then clipped onto a glass slide via a simple latching system (Fig. 3b) ensuring simple calibration. Flow patterning is achieved by dynamically controlling the flow streams, effectively creating a reconfigurable streaming display of spatially segregated reagents capable of processing several small surfaces with chemicals in parallel rather than in series as per the scanning probe mode.

A library of 3D-printed MFM heads, each corresponding to a different geometry obtained by a conformal transformation of the dipole (Table 1), were fabricated. The MFMs.stl files are available in the supplementary material. Figure 3c–h presents six side-by-side comparisons between experimental results and theory. A near perfect correspondence between the experimental and the theoretical model was found for all cases, validating

**Table 1 Examples of simple transforms that can be applied to the dipole solution to obtain new geometries**

| | |
|---|---|
| Straight flow | $\omega = \log(z) - \alpha \log(z+1)$ |
| Microfluidic quadrupole | $\omega = (2z+1)^{1/2}$ |
| Flower multipole | $\omega = (z+1)^{1/n}$ |
| Polygonal multipole | $\omega = z^{1/n}$ |
| Impinging flows | $\omega = 1/z - 1$ |

A more exhaustive table (Supplementary Table 3) is presented in the SI

the advection-diffusion model for both known and new geometries. Further comparisons are available in Supplementary Figs. 5 and 6.

**Reconfigurable microfluidic multipole devices.** Conformal mapping can be used to map any known MFM pattern onto arbitrarily complex geometries comprising an unlimited number of apertures, and to position the apertures according to the predictions of the model. These patterns can then be fabricated by simply positioning the apertures where dictated by the new map. The number of different ways to assign $n+2$ flow modes (aspiration, stop, or injection of $n$ different reagent conditions) for a MFM device with $a$ apertures is large, scaling with $(n+2)^a$. Several symmetries can be exploited in laying out the position of the apertures to achieve periodic patterning of a surface in order to achieve particular confinement geometries. In this section, we highlight two of them, the rotationally-symmetric MFM (rMFM) and the translationally-symmetric MFM (tMFM), obtained via our theory-assisted design approach.

The rotationally-symmetric MFM configuration is achieved by making all aspiration apertures in a MFM superpose to form a central drain around which injection apertures are placed to form the vertices of a regular polygon. The advection-diffusion profile can thus be expressed exactly by a transformation of the type "flower" presented in Table 1 and already described in Figs. 2c, f, and 3f, g. rMFMs allow the largest theoretical number of possible independent confined reagent conditions ($a-1$) using $a$ apertures (see Fig. 4a). Furthermore, the confinement area (henceforth defined as "petals" to extend the flower analogy) can easily be kept stable by compensating for the flow variations while some openings are turned on and off. For that reason, rMFM can be used as a chemical stroboscope, enabling the precise and independent spatiotemporal control of chemical pulses above a surface.

Figure 4a shows an experiment where the amplitude (given by the reagent concentration), the frequency, and the duty cycle of several chemical pulses are controlled independently. One petal of the rMFM is always exposed to the reagent, one is never exposed, and the remaining six petals are exposed to 3 different frequencies (period of 12 s, 16 s, and 24 s) with 2 different duty cycles (25% and 50%) (Fig. 4b). rMFMs demonstrated flawless control with characteristic times to achieve steady state between pattern changes under 1 s.

Petal shapes are affected by the number of apertures turned on and off at any given time. However, this effect can be effectively compensated by tuning in real time the flow rate in each of the apertures according to the exact flow model described above, yielding fixed-size petals and thus independent confinement areas (see Supplementary Note 4). The real-time adjustment of several petals leads to a rich set of dynamic applications for rMFM (Supplementary Fig. 7 and Supplementary Videos 1–3).

Another natural way to place openings in a MFM device is to lay them in a periodic array with translational symmetry, which yield what we define a translationally-symmetric MFM. This

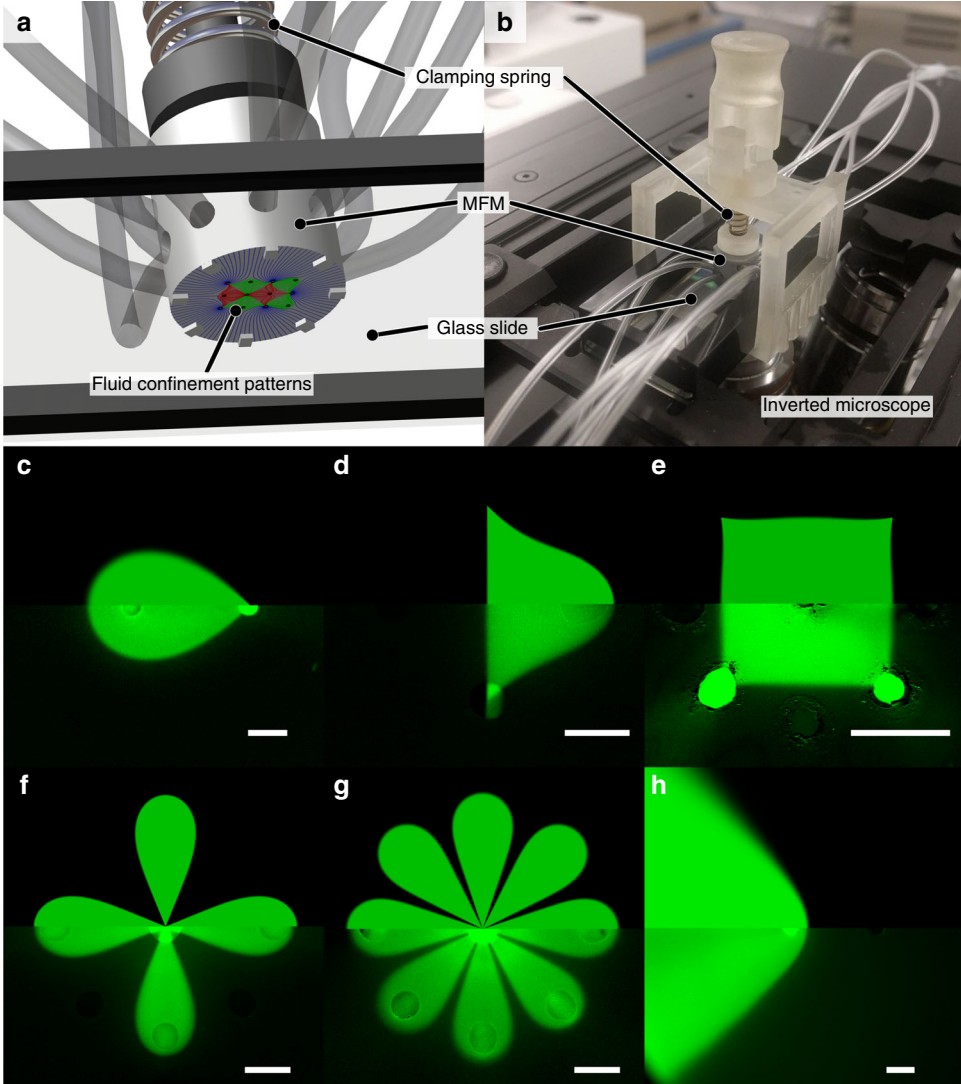

**Fig. 3** Experimental setup and side-by-side comparison between theory and experiments. **a** Schematics of a fixed MFM setup. The MFM is precisely positioned over the surface with a gap controlled by the spacers. **b** Picture of the experimental setup with holder and MFM clamped atop of an inverted microscope. **c–f** Side-by-side comparison between analytical and experimental results for various multipolar configurations. The top half of each subfigure is the theoretical concentration profile while the bottom half is a micrograph of a fluorescent dye injected using the MFM (Pe ~$10^2$, Reynolds number ~$10^{-3}$). **c** Microfluidic dipole. **d** Microfluidic quadrupole. **e** Polygonal multipole (**f**) 4-petal axisymmetric "flower" multipole (**g**) 8-petal axisymmetric "flower" multipole (**h**) Asymmetric impinging flows of different concentrations. Scale bars = 500 μm

configuration relies on the modulation of several injection/ aspiration flow apertures to spatially address, in parallel, multiple regions with multiple reagents in a dynamic fashion. tMFMs offer a significant advantage over rMFM: they increase the number of achievable reagent patterns at the cost of introducing more aspiration apertures. As a proof of the versatility of the technology, a 12-aperture rectangular tMFM was used to make 28 confinements patterns in a single experiment lasting less than two minutes (Fig. 4c and Supplementary Video 4). The system showed fast transient times of around 1 s. Depending on the injection and aspiration configurations chosen, different sets of fluid patterns can be made (Supplementary Fig. 8) presents a selection of the possible configurations). To give an order of magnitude, for a 12-aperture tMFM with 2 different reagents injection, there are $1.6 \times 10^7$ ($4^{12}$) different possible configurations. It can be noted that every MFP flow pattern using a point source design published until now can be made with a single tMFM with the appropriate injection/aspiration configuration, with the exception of the circular probe[36].

**Immunofluorescence assay using microfluidic multipoles**. To showcase the applicability of MFM for multistep long-lasting surface patterning applications, a fully automated three-step immunoassay was carried out (Fig. 5). Working on functionalized slides with spots of immobilized goat IgG anti-mouse antibodies, a staggered 12-aperture tMFM device was used to incubate 6 different concentrations of antigen (mouse anti-human IgA heavy chain) using the central 7 apertures to form a 6-sided rMFM subset within it (Fig. 5a). After an incubation period of 50 min, the antigen injection apertures were stopped, and the 4 corner apertures were used to inject the fluorescently-labeled detection antibody (Donkey anti-Mouse IgG) over the previously exposed antigen zone for 1 h (Fig. 5b). At the end of the detection antibody incubation time, the injections were stopped for 10 s to aspirate the detection antibody between the tMFM and the surface. The central aspiration was then turned off, and the 12th aperture was used to inject the washing buffer for 15 min (Fig. 5c). Following retrieval of the slide, rinsing, and drying under a stream of nitrogen, it was imaged immediately.

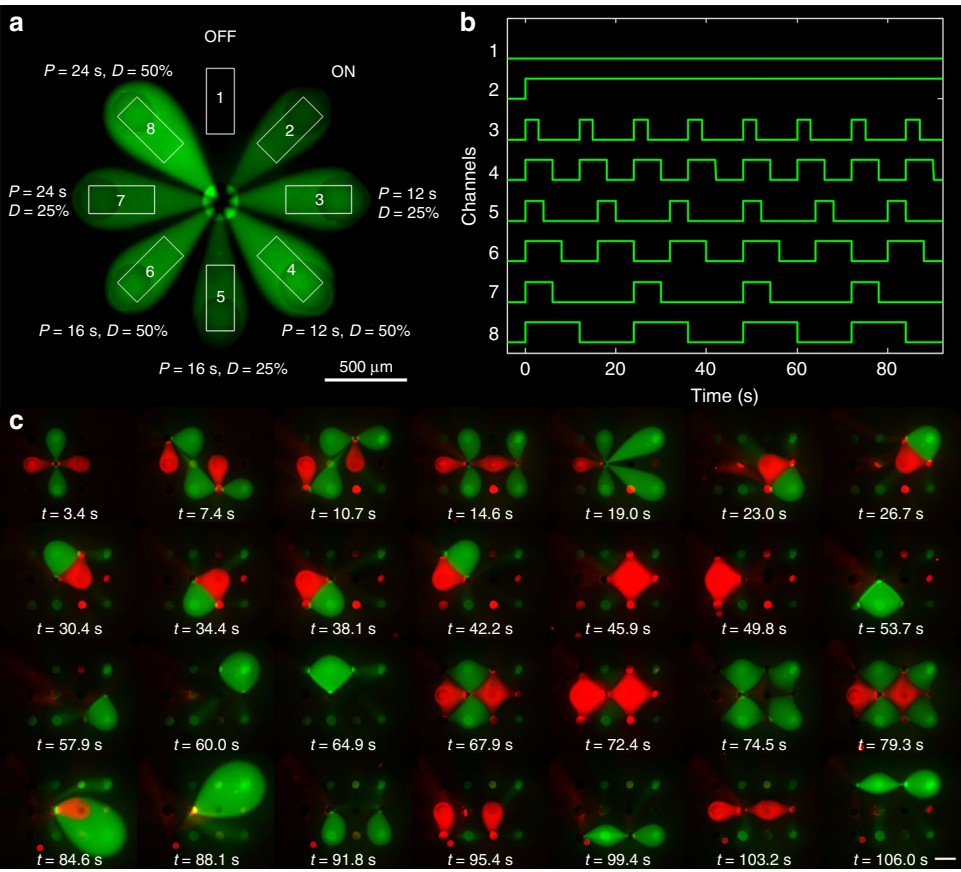

**Fig. 4** Microfluidic multipole devices. **a** Fluorescence micrograph showing the confinement pattern of a rMFM device. **b** Graph representing the periodic exposure to reagents for each confinement area of a rMFM used as a chemical stroboscope. **c** Fluorescence micrographs showing 28 different confinement patterns made with a 12-aperture tMFM during a single experiment lasting less than 2 min. Scale bar represents 500 μm

The fluorescent signal as function of antigen concentration was used to calculate the binding curves of the assay, as shown in Fig. 5d, e. The area not exposed to antigens and stained by detection antibody was considered as a control area and used to calculate the background signal. Confinement areas are well defined and show no sign of cross-talk between them. Previous experiments showed that, as expected, a confinement area with $[Ag] = 0$ gives similar results to the detection antibody stained background (see Supplementary Fig. 9). A limit of detection (LOD) of 13 pM/ml (~2 pg/ml) was obtained, which is close to the best LODs that can be obtained using the sandwich assay format and common Enzyme-linked immunosorbent assays (ELISA)[37].

## Discussion
From a theoretical standpoint, MFMs are a general theoretical concept to guide open space microfluidic design. We have provided in this paper a complete formalism to study advection-diffusion of reagents in 2D open-space microfluidics. The analytical solutions provided are asymptotically accurate for high Péclet numbers (Pe > 10 in practice), which encompasses most applications of open-space microfluidics. The method significantly expands the microfluidics toolbox by identifying new, broad classes of multipolar flows and concentration patterns with known analytic expressions and by providing a simple strategy for theory-guided design of open-space microfluidics systems. Incidentally, it also yields solutions for a class of unexplored diffusion problems in 2D laminar analogs to impinging jets[15,38] relevant to microfluidics. Given the versatility of the method and the infinite possibilities to map multipolar flows onto new geometries

involving reagent patterning, the theoretical approach described effectively unifies previously published MFP flow patterns under the more general concept of MFMs of which scanning MFPs are but a subset of the whole possible "design library".

It is also worth emphasizing that conformal invariance is a property of the advection-diffusion equation in 2D potential flows that can be used regardless of whether we know or not the exact solution to a specific problem. Any initial 2D image of the diffusion profile can be mapped using conformal transforms, be it analytical, numerical, or experimental. Thus, transport profiles generated numerically or observed experimentally can also be used as an initial "known solution" onto which conformal transforms are applied to yield new solutions without solving any differential equations (see examples in Supplementary Note 5). Conformal mapping is therefore versatile and useful experimentally even in cases where no a priori analysis has been made.

From a technological standpoint, MFMs are a step towards multipolar open-space microfluidic technologies. They improve on other open-space microfluidic systems on two fundamental points: reconfigurability and parallelization. Hence, fixed spatio-temporally reconfigurable MFM devices could conveniently replace scanning MFPs for various surface processing applications and potentially channel-based microfluidics for some special cases. They can also handle effectively, in a parallel fashion, long (minutes to days) incubation times typical of immunohisto-chemical (IHC) staining[12], immunoassays, and DNA hybridization assays. The simplified and smaller positioning system involving spacers and surface latching mechanism makes for a simple integration of fixed MFM devices inside an incubator, which can be required for long experiments such as

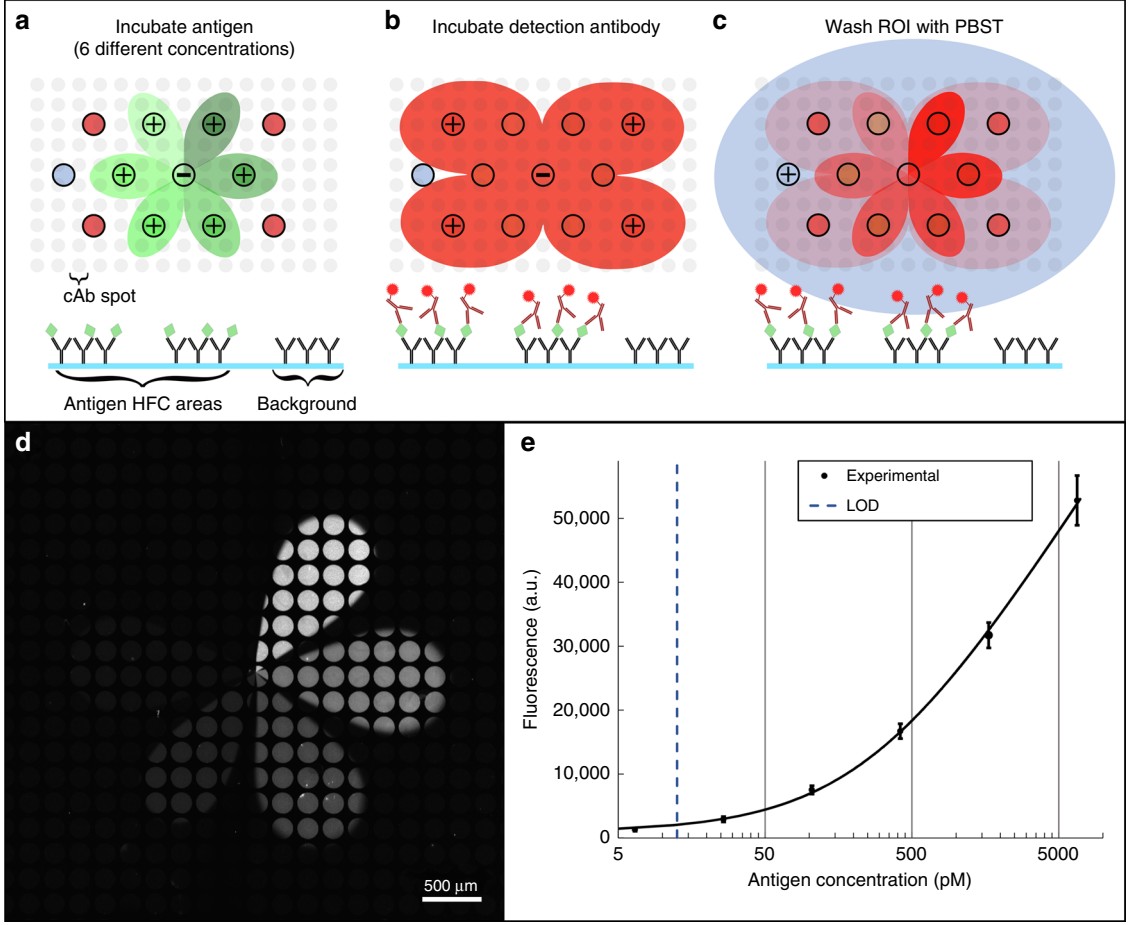

**Fig. 5** Immunofluorescence assay experiment using a microfluidic multipole. **a–c** Workflow of an immunofluorescence assay using a staggered tMFM. **a** The device is used as a 6-sided rMFM device to expose the capture antibody to 6 different concentrations of antigen. **b** The corner apertures are then used to expose the previous areas with detection antibodies. **c** The last aperture is used to wash the slide. **d** Micrograph of the detection antibody of the immunoassay made with the tMFM device. **e** Experimental binding curve of the immunoassay. Error bars represent the standard deviation over 6 spots

immunofluorescence staining. In all assays, MFM also possess the obvious advantage that control and experimental conditions are tested within at most a few millimeters of each other all the while being completely isolated biochemically from each other.

Combining both the ability to pattern entire open surfaces and to manipulate streams of fluids in geometrically well-defined structures, fixed MFM devices effectively bridge the gap between mobile MFPs and closed-channel microfluidics. They afford spatiotemporal control with a resolution dictated by the spacing and number of apertures, and with combinations that scale along with the number of apertures as well. Finally, through the accurate theoretical framework proposed, their transport behavior is now as predictable as that of closed-channel microfluidics systems.

In this article, MFM devices of up to 12 apertures were 3D printed and operated using syringe pumps as a proof of concept of their operation. It is understood that MFMs with even more apertures would allow the generation of more complex patterns but scaling up of the current setup is a challenge that remains to be solved because the flow in each aperture is currently controlled by an individual syringe pump. However, several available strategies, using fluidic routing systems[39], multilayer soft lithography valves arrays[40], or upstream gradient generators[41], could provide avenues to form large MFMs with compact control systems, thus enabling random access surface processing and gradient formation onto very large areas. The current resolution of commercial 3D printing technologies limits the integration of large numbers

of connectors and multiplexers. However, custom 3D printer and resin[42] have been made demonstrating the integration of 88 connectors per mm², with a feature resolution <20 μm[43,44], which could allow making MFM with higher resolution and higher density applications, and serve many more applications in the life sciences.

In the future, MFM devices may play an important role to study spatially resolved systems with fast transient kinetics. Using the presented advection-diffusion models, precise biochemical gradients could be generated, making MFM devices potentially useful to study biological processes sensitive to gradients, like neutrophils migration through chemotaxis[45], stem cells differentiation[46], and neuronal developement[47,48].

Finally, we hope that the proposed framework to study hydrodynamics and diffusion in multipolar flows will spark interest in open-space microfluidics much the same way rigorous quasi-1D models contributed to the development of channel-based microfluidics devices, and inspire fundamentally new applications exploiting the unique properties of these planar flows.

## Method

**Conformal mapping.** All images were generated from analytical expressions using Matlab R2016a (MathWorks, Inc., Natick, USA). No numerical simulations were used in this article.

**Microfluidic multipoles design and fabrication.** MFMs were designed using Catia V5 (Dassault Système, Vélizy-Villacoublay, France) and then 3D printed

using a 27 μm resolution stereolithography printer (Freeform Pico and Pico 2 HD, Asiga, Alexandria, Australia). Plasgrey V2 resin (Asiga) and Pro3dure GR-1 resin (Pro3dure medical, Dortmund, Germany) were used to print the MFMs. Plasgrey V2 MFMs were printed using the Freeform Pico printer with a wavelength of 405 nm and an exposure time of 1.150 s. GR-1 MFMs were printed using the Pico 2 HD printer with a wavelength of 385 nm and an exposure time of 0.468 s. A layer thickness of 50 μm were used for both resins. After the printing, they were cleaned in an isopropanol bath in a sonicator (Branson, Danburry, USA). 1/16" O.D. Tygon tubes (Cole Parmer, Vernon Hills, USA) were then plugged and glued to the MFMs using cyanoacrylate glue. The fabrication method is further described in a previously published article[23]. MFMs with auto-alignment pillars required no active component and were simply clamped on a glass slide using a simple 3D printed setup and a spring. Commercially available (ProPlate® 1 Well Slide Module, Grace Bio-Labs, Bend, USA) and custom-made microscope slide walls were used depending on the experiment. Flow rates were controlled by NEMESYS syringe pumps (CETONI, Korbußen, Germany). The pump system was controlled by a custom-made LabView code (National Instrument, Austin, USA).

**Experimental characterization**. Fluids confinement areas were imaged using on inverted epifluorecence microscope (Axio Observer.Z1, Zeiss, Oberkochen, Germany) with the Lavision sCMOS camera (Göttingen, Germany). Fluorescein sodium salt (Sigma Aldrich, Saint-Louis, USA) dissolved in ultrapure water was used as primary fluorophore for the experiment. For experiment requiring a second fluorophore, a solution of Propidium Iodide (Sigma Aldrich) and DNA sodium salt (DNA sodium salt from salmon testes, Sigma Aldrich) dissolved in ultrapure water was used. For each channel, the background was subtracted, and image intensities amplified using MATLAB before being merged. The background used for background removal was an image of the probe with the injection and aspiration apertures stopped. Videos were made using the same method frame-by-frame and then compressed using MATLAB.

**Immunofluorescence assay experimental procedures**. Glass Slides functionalized with 2D Aldehyde were purchased from PolyAn GmbH (Berlin, Germany). Goat anti-Mouse IgG (H + L) labeled with Alexa Fluor 488 (Thermo fisher scientific, Waltham, USA) was used as capture antibody and was spotted using an inkjet spotter (sciFLEXARRAYER SX, Scienion, Berlin, Germany) in a printing buffer (100 μg/mL cAb, 15% 2,3-butanediol and 15% betaine in PBS). Slides were incubated overnight at 70% humidity and then blocked for 2 hours in PBS buffer with 3% BSA and 0.1% Tween 20. Both the antigen (Mouse anti-Human IgA (Heavy chain), Thermofisher scientific) and the detection antibody (Donkey anti-Mouse IgG (H + L) tagged with Alexa Fluor 647, Thermofisher) were diluted in PBS buffer with 3% BSA and 0.1% Tween 20. The 6 antigen concentration points were prepared by performing a four-fold dilution series with a starting concentration of 1 μg/mL. Antigens were incubated for 50 min on the slide using a staggered tMFM with $\alpha = 1.3$, $Q_{inj} = 0.3$ μL/s and a gap of 50 μm. The detection antibody was subsequently incubated for 60 min with $\alpha = 1.05$ and $Q_{inj} = 0.3$ μL/s. The slide was washed with the MFM by injecting 3 μL/s of PBST (PBS and 0.1% Tween 20) with one aperture for 15 min. The MFM was then removed, and slides were quickly dipped in ultrapure water before being dried under a nitrogen stream. Results were imaged using a fluorescence microarray scanner (Innoscan 1100 AL, Innopsys, Carbonne, France) at 635 nm. The binding curve was made by calculating the means and standard deviation for 6 antibody spots for each concentration. The curve was fitted with a 5 parameters logistic curve using MATLAB. Limit of detection (LOD) was defined as:[49] LOD = mean_blank + 1.645(SD_blank) + 1.645(SD_lowest concentration sample).

## Data availability

The authors declare that the data supporting the findings of this study are available within the article and its Supplementary information files.

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

## Acknowledgements

P.A.G. and E.B. acknowledges a graduate fellowship from the Fond de Recherche du Québec Nature et Technologies. T.G. acknowledges funding from the Fonds de Recherche du Québec (FRQ), "Établissement de nouveaux chercheurs" and "Équipe" programs, and the National Science and Engineering Research Council of Canada (NSERC – RGPIN - 06409). We thank the Canadian Microsystems Center (cmc.ca) for access to a shared infrastructure. This research was conducted, in part, thanks to support from the Canada First Research Excellence Fund (TransMedTech Institute). Special thanks to Cetoni GmbH for support with the pump system. We also thank Suzanne Gaudet (HMS/Dana-Farber) for useful discussions.

## Author contributions

E.B. and T.G. designed the theoretical part of the research. E.B., G.L. and T.G. carried out the analytical modeling. P.A.G. and T.G. designed the experimental part of the research. P.A.G. designed and fabricated the MFMs and experimental setup and carried out the fluorescence experiments. P.A.G. and E.B. analyzed the results. P.A.G., F.N., D.J. and T.G. designed the immunoassay experiments. P.A.G. and F.N. realized the immunoassay experiments. P.A.G., E.B. and T.G. wrote the manuscript. All authors reviewed the manuscript.

## Additional information

**Competing interests:** T.G., P.A.G., and E.B. declare a competing financial interest under the form of a patent pending in the US. The patent application covers the rMFM and the tMFM geometries of microfluidic multipoles. Patent application number 62739685. The remaining authors declare no competing interests.

