## [peer review file · Nature Communications]

Reviewer #1 (Remarks to the Author):

From a fluid mechanics perspective I found this paper an interesting and important application of Bazant's relatively recent work on the conformal invariance of the steady 2D advection-diffusion equation for a tracer c . Several new advances are made which bring together theory and experiment: (i) the derivation of exact solutions for the tracer distribution driven by sources and sinks (here, termed the aspirator) distribution; (ii) development of new MFM devices based on (i); and (iii), practical demonstration of the implementation of these devices. The visual comparison between theory and experiment shown in Fig. 3 is convincing. The ability of conformal mapping to generate further flows (multipolar etc.) is exploited both theoretically and practically.

While I am not familiar with the applications, I was nevertheless very interested to read about them. I am sure other readers of this journal would be equally interested and I am keen to see this work published especially given its potential impact to microfluidics. I do, however, have some questions about the theory which the authors should address (see below) before I can recommend publication. For the most part the paper is well written and relevant works are nicely summarised, but I include a few minor remarks also below.

Questions:

Key to the derivation of the tracer distribution is the mapping of the source sink problem in the z -plane to the potential or hodograph (ϕ, ψ) -plane and then exploiting the exact solution given by, for example, Bazant (2004). I am unclear about the following aspects of this key step:

(a) Why is $c=1/2$ on the separating streamline? Indeed, why should c be constant at all on this streamline. Some more detail on this point needs to be provided in addition to the lines 185-187.

(b) In the hodograph mapping procedure, the curved portion of the bounding streamline from z_{stag} to the aspirator ($z=-1$) in the upper half of the z -plane is mapped to the semi-infinite straight line (along which $\psi=const$ which can be taken to be zero without loss of generality) from ϕ_{stag} to $\phi=\infty$ at the aspirator (note: including a figure showing which regions and curves of the z -plane map to their counterparts in the (ϕ, ψ) -plane would be very helpful) Now, what then, about the portion of the same bounding streamline (i.e. $\psi=0$) along the real axis from $z=-1$ to $z=z_{stag}$ which goes through the source at $z=0$ where $\phi=\infty$ again? Where does this straight line segment map to in the (ϕ, ψ) -plane?

(c) It is argued (lines 206-215) that the the choice of sign in (6) depends on whether the region exterior or interior to the bounding streamline in being considered. This is not obvious to me. More precisely, for questions similar to (b) above it is not clear why the exterior region also maps to the same region in the (ϕ, ψ) -plane. For example, where does the portion of the straight

streamline (along which $\psi=0$) from z_{stag} along the real z -axis to $Re(z)=\infty$ map to? And the portion of the streamline ($\psi=0$) along the real axis from $Re(z)=-\infty$ to $z=-1$?

(d) In S1's derivation of the polar equation for the bounding streamline, note that there is a restriction on θ namely $\theta_1 < \pi/\alpha$ (otherwise r can take on negative values which is obviously not permitted)

The visual comparison between theory and experiment shown in Fig. 3 is remarkable. It would be even better would be a quantitative comparison and justify the phrase 'near perfect' (line 291) e.g. can theoretical and experimental isolines of $c(x,y)$ also be given?

Minor remarks

Lines 45-47 give some remarks about the flow being restricted to 2D. How this is achieved is still not clear to me. A Hele-Shaw cell confines flow to a thin gap between upper and lower boundaries (or plates). What is the upper boundary (plate) in the present work? Is the upper boundary (whatever it is) sufficiently large to ensure all the dynamics is 2D?

There is net loss of tracer since $\alpha < 1$, so at first glance it seems surprising that the distribution of c is steady. It is probably worth explicitly noting that steadiness is a consequence of diffusion balancing advection.

In table 1 the map $\omega=1/z$ is mentioned a generating 'impinging flows'. This map send the source to infinity while leaving the sink (aspirator) unchanged. The resulting streamlines are then radial lines emanating from $z=-1$, and presumably the bounding streamline is a circle centred at $z=-1$. In what sense is this 'impinging'?

Reviewer #2 (Remarks to the Author):

This paper demonstrates that microfluidic multipoles (MFM), do indeed confirm mathematical models, when compared directly with experimental results. The authors demonstrated that theory

can be visualised, practically, using specifically designed 3D printed MFM devices, generating patterns with devices of up to 12 apertures. Finally, an immunoassay with 3-steps was demonstrated resulting in comparable performance to typical ELISA assays, with an acceptable margin of error.

The use of 3D printing for a specialised method (in this case MFM) and validating with theory, not only makes future research into MFM systems more accessible, but also strengthens 3D printed microfluidics as a useful tool for the scientific community.

At present the paper does not have significant flaws that would impede publication. However, there are aspects of the introduction, and figures that need to be addressed; particularly, the lack of literature regarding 3D printed microfluidic devices, in the introduction.

Overall, the approach made by the authors in this work is logical, and supported by high quality data, and figures. Particularly, the included SI videos of the various configurations demonstrate the operation of the device(s) in a clear manner.

However, there are some details excluded in the Methods concerning the fabrication, and characterisation of the 3D printed MFM devices. It is appreciated that the authors have filed for patent, but at present the practical work could be difficult to reproduce.

Error bars are appropriately described in Fig 5(c), and in SI – Fig 5; all other data is not graph based.

Minor aspects of the paper that should be addressed:

- Fig 1 – while correspondence between the theoretical streamlines, and micrographs of experimental results are excellent in (d), there is an inconsistency. Here, the interface line bulges from Q6(t) to Q7(t), could this be due to errors in the 3D printed aperture? A comment on these artefacts, would be of benefit to the work.
- SI – Fig 4 – Is it perhaps a similar effect that is seen on the ‘line’ pattern? (see previous)
- Methods - Exposure times for the fabrication.
- Methods – Why were two materials used for making the devices?
- SI - .stl models of the MFM devices used in the work could be included in the SI
- Fig 3 (a-b) – this figure could benefit from being moved to an earlier point in the manuscript. In addition, simple labels for the CAD render and/or in the figure caption, would greatly enhance the figure.
- Fig 3 (c-h) – Considering that half of the experiment is covered in the figure at present, it is difficult to draw a full conclusion on the overlap of the MATLAB image. Perhaps it could be improved further by separating the theory, and experimental (as in Fig 1), but also including a full overlay of the two patterns?

At present to the knowledge of the reviewer, the manuscript has only one reference to a 3D printed (3DP) microfluidic device. Considering that the conclusions result from direct comparison of theory with a 3DP microfluidic device, it would be pertinent to include a brief acknowledgement of advancement in that field, and how that 3DP is of benefit to the work described in the manuscript. There are numerous reviews on the subject (perhaps too many), but a number have been well received by the community.

All the main points in the abstract, introduction, and conclusion are well outlined, except for the missing aspects mentioned previously about 3DP microfluidics.

The derivation of the mathematical equations for microfluidic multipoles is not within the reviewers current expertise; however, the mathematics themselves can be followed.

Reviewer #1 (Remarks to the Author):

From a fluid mechanics perspective I found this paper an interesting and important application of Bazant's relatively recent work on the conformal invariance of the steady 2D advection-diffusion equation for a tracer c . Several new advances are made which bring together theory and experiment: (i) the derivation of exact solutions for the tracer distribution driven by sources and sinks (here, termed the aspirator) distribution; (ii) development of new MFM devices based on (i); and (iii), practical demonstration of the implementation of these devices. The visual comparison between theory and experiment shown in Fig. 3 is convincing. The ability of conformal mapping to generate further flows (multipolar etc.) is exploited both theoretically and practically.

While I am not familiar with the applications, I was nevertheless very interested to read about them. I am sure other readers of this journal would be equally interested and I am keen to see this work published especially given its potential impact to microfluidics. I do, however, have some questions about the theory which the authors should address (see below) before I can recommend publication. For the most part the paper is well written and relevant works are nicely summarised, but I include a few minor remarks also below.

[Authors]: We thank the reviewer for the encouraging words and sharing her/his interest for the proposed applications.

[Reviewer 1]

Questions:

Key to the derivation of the tracer distribution is the mapping of the source sink problem in the z -plane to the potential or hodograph (ϕ, ψ)-plane and then exploiting the exact solution given by, for example, Bazant (2004). I am unclear about the following aspects of this key step:

(a) Why is $c=1/2$ on the separating streamline? Indeed, why should c be constant at all on this streamline. Some more detail on this point needs to be provided in addition to the lines 185-187.

[Authors] It is indeed not immediately obvious why the streamline going through the stagnation point should have a constant concentration, or why this concentration should be $1/2$. In the case of the microfluidic dipole (and by extension any geometry obtained by transforming the dipole solution), this fact can be seen by transforming the problem not to streamline coordinate, but rather to a purely extensional flow (achieved by taking the square root of the hodograph map). In this geometry, the inlet of concentration 1 maps to the limit $y \rightarrow \infty$, while the "inlet" at infinity ($r \rightarrow \infty$ in the original geometry), maps to the limit $y \rightarrow -\infty$.

The transport problem then becomes convection-diffusion in a purely extensional flow with fixed concentration at $y = \pm \infty$. This problem is symmetrical along the real axis and we can see by symmetry that the axis $y = 0$ has concentration $c = 1/2$ irrespectively of the ratio of aspiration to injection flow rates (α) or the properties of the tracer. This axis is the one that maps to the separating line.

The reason we chose to solve the problem in streamline coordinates and not this purely extensional flow (while they are both equivalent for the dipole solution) is that the streamline coordinates are a more natural geometry for solving more complex problem in which there could be several lines of fixed concentration (like in the case of the asymmetrical quadrupole or nested dipole configuration).

Action Taken: Added supplementary note “Supplementary Note 1: Concentration on the separating line” clarifying this point. We added a sentence referring to this supplementary note “The concentration at the stagnation point, as well as on the segment of streamline connecting the stagnation point to the aspiration aperture, is necessarily $c = 1/2$, which can be seen by mapping to a purely extensional flow and recognizing the problem’s symmetry in this domain (see Supplementary Note 1).”

(b) In the hodograph mapping procedure, the curved portion of the bounding streamline from z_{stag} to the aspirator ($z=-1$) in the upper half of the z -plane is mapped to the semi-infinite straight line (along which $\psi=const$ which can be taken to be zero without loss of generality) from ϕ_{stag} to $\phi=\infty$ at the aspirator (note: including a figure showing which regions and curves of the z -plane map to their counterparts in the (ϕ, ψ) -plane would be very helpful) Now, what then, about the portion of the same bounding streamline (i.e. $\psi=0$) along the real axis from $z=-1$ to $z=z_{stag}$ which goes through the source at $z=0$ where $\phi=\infty$ again? Where does this straight line segment map to in the (ϕ, ψ) -plane?

[Authors] The real axis $y = 0$ in the dipolar flow domain actually maps to 4 different regions in the streamline coordinate plane, the separation of these regions being the two singularities and the stagnation point.

The image of each of these segments can be found by using the definition of ψ

$$\psi = \text{atan}(y/x) - \alpha * \text{atan}(y/(x+1))$$

Which is just the sum of the angle from the origin and $-\alpha$ times the angle from the aspiration.

The region of the real axis from $x = -\infty$ to the aspiration aperture at $z = -1$ maps to the horizontal axis $\psi = (1 - \alpha)*\pi$

The region between the aspiration and injection aperture maps to the horizontal axis $\psi = \pi$.

Finally, the segment between the origin and the stagnation point as well as the segment between the stagnation point and $x = +\infty$ maps to two sides of the semi-infinite segment $\psi = 0$, ϕ goes from $-\infty$ to ϕ_{stag}

A table was added to the supplementary material containing the position of each segment and where it maps in the streamline coordinate plane

Action Taken: Added supplementary table 2: Features of the dipole flow and their image in streamline coordinates.

Supplementary Table 2: Features of the dipole flow and their image in streamline coordinates

Z-plane coordinates	Streamline coordinates
Separating line (See supplementary note 2 for expression)	$\phi \in]\phi_{stag}, \infty[$ $\psi = 0$
$x \in]-\infty, -1[$ $y = 0$	$\phi \in]-\infty, \infty[$ $\psi = (1 - \alpha)\pi$
$x \in]-1, 0[$ $y = 0$	$\phi \in]-\infty, \infty[$ $\psi = \pi$
$x \in]0, x_{stag}[$ $y = 0$	$\phi \in]-\infty, \phi_{stag}[$ $\psi = 0^+$
$x \in]x_{stag}, \infty[$ $y = 0$	$\phi \in]-\infty, \phi_{stag}[$ $\psi = 0^-$

(c) It is argued (lines 206-215) that the the choice of sign in (6) depends on whether the region exterior or interior to the bounding streamline in being considered. This is not obvious to me. More precisely, for questions similar to (b) above it is not clear why the exterior region also maps to the same region in the (ϕ, ψ) -plane. For example, where does the portion of the straight streamline (along which $\psi=0$) from z_{stag} along the real z -axis to $Re(z)=\infty$ map to? And the portion of the streamline ($\psi=0$) along the real axis from $Re(z)=-\infty$ to $z=-1$?

[Authors] We believe the answer to the previous section, as well as the added supplementary table, should answer this question as well.

Action Taken: Added supplementary table 2: Features of the dipole flow and their image in streamline coordinates.

(d) In SI's derivation of the polar equation for the bounding streamline, note that there is a restriction on θ namely $\theta_1 < \pi / \alpha$ (otherwise r can take on negative values which is obviously not permitted)

[Authors] Actually we only take the θ to be between $-\pi$ and π , otherwise we get streamlines that we are not interested in, way before we hit the limit $\theta = \pi * \alpha$. This was clarified in the SI by specifying the range of θ used

Action Taken: Modified “Supplementary Note 2” by specifying the range of θ used.

The visual comparison between theory and experiment shown in Fig. 3 is remarkable. It would be even better would be a quantitative comparison and justify the phrase 'near perfect' (line 291) e.g. can theoretical and experimental isolines of $c(x,y)$ also be given?

[Authors] We thank the reviewer for his appreciation of our represented data. We have searched for a method of accurately comparing experimental and theoretical concentration profiles and obtaining a quantitative (numerical) result. It has proven challenging as most methods produce either inconclusive, overly positive or overly negative results. To make the comparison clearer than the visual side-by-side, we added to the supplementary figures a graph with cross-sections of the concentration profile.

Action Taken: Added supplementary Figure 3 comparing concentration profile along several different cross sections

Supplementary Figure 3 Comparison between theoretical model and experimental results for different cross-sections of a microfluidic dipole. How each cross-section maps to the experimental image is illustrated in a), while b)-e) show the comparison of experimental and theoretical results. It is to be noted that the discrepancy in subfigure e) is due to the presence of another aperture on the multipole, which hides a part of the concentration gradient (visible in subfigure a). c and r in the graphs are in adimensional units. Scale bar: 300 μm .

Minor remarks

Lines 45-47 give some remarks about the flow being restricted to 2D. How this is achieved is still not clear to me. A Hele-Shaw cell confines flow to a thin gap between upper and lower boundaries (or plates). What is the upper boundary (plate) in the present work? Is the upper boundary (whatever it is) sufficiently large to ensure all the dynamics is 2D?

[Authors] The head of the microfluidic multipole constitutes the “top plate” in this flow. The flat part of the head has a radius of about 1 cm while the area in which the flow of interest is generated is of the order of 2 or 3 mm, ensuring enough space around to justify the flow being considered between two “semi-infinite plates”.

Action Taken: Slightly modified sentence at line 137 for more clarity: “We assume an infinite flow domain, which in practice is realized by having the surface of the injection device be a few times larger than the maximum interaperture distance.”

There is net loss of tracer since $\alpha < 1$, so at first glance it seems surprising that the distribution of c is steady. It is probably worth explicitly noting that steadiness is a consequence of diffusion balancing advection.

[Authors] This is a good point, we will add it to the text

Action Taken: Added sentence to line 139 of the text: In cases where $\alpha > 1$, there exists a steady-state where advection balances diffusion, allowing us to use the steady advection-diffusion equation

In table 1 the map $\omega = 1/z$ is mentioned a generating ‘impinging flows’. This map send the source to infinity while leaving the sink (aspirator) unchanged. The resulting streamlines are then radial lines emanating from $z = -1$, and presumably the bounding streamline is a circle centred at $z = -1$. In what sense is this ‘impinging’?

[Authors] There was actually an error in the table, and we thank the reviewer for finding it. The correct transform should be $\omega = 1/z - 1$, which maps sends the aspiration aperture to infinity (and not the injection). This has been corrected in the revised version

Action Taken: Corrected Table 1, the last line should now read “ $\omega = 1/z - 1$ ” and not “ $\omega = 1/z$ ”

Reviewer #2 (Remarks to the Author):

This paper demonstrates that microfluidic mutlipoles (MFM), do indeed confirm mathematical models, when compared directly with experimental results. The authors demonstrated that theory can be visualised, practically, using specifically designed 3D printed MFM devices, generating patterns with devices of up to 12 apertures.

Finally, an immunoassay with 3-steps was demonstrated resulting in comparable performance to typical ELISA assays, with an acceptable margin of error.

The use of 3D printing for a specialised method (in this case MFM) and validating with theory, not only makes future research into MFM systems more accessible, but also strengthens 3D printed microfluidics as a useful tool for the scientific community.

At present the paper does not have significant flaws that would impede publication. However, there are aspects of the introduction, and figures that need to be addressed; particularly, the lack of literature regarding 3D printed microfluidic devices, in the introduction.

[Authors] We thank the reviewer for her/his appreciation of our work and kindly refer her/him to the final comment below for a complete response to the constructive comment on 3D printing.

Overall, the approach made by the authors in this work is logical, and supported by high quality data, and figures. Particularly, the included SI videos of the various configurations demonstrate the operation of the device(s) in a clear manner.

However, there are some details excluded in the Methods concerning the fabrication, and characterisation of the 3D printed MFM devices. It is appreciated that the authors have filed for patent, but at present the practical work could be difficult to reproduce.

Error bars are appropriately described in Fig 5(c), and in SI – Fig 5; all other data is not graph based.

Minor aspects of the paper that should be addressed:

- Fig 1 – while correspondence between the theoretical streamlines, and micrographs of experimental results are excellent in (d), there is an inconsistency. Here, the interface line bulges from Q6(t) to Q7(t), could this be due to errors in the 3D printed aperture? A comment on these artefacts, would be of benefit to the work.

[Authors]

If we understand the question well, the reviewer is asking why the red “square” pattern line are deformed and are not symmetric relatively to a vertical axis passing by Q7(t) and suggests that it may be due to 3D printing artefacts. In fact, the aperture size and shape has a negligible effect on the obtained patterns (except when really close to the apertures), as long as they are relatively small compared to the patterns that are being made. We have extensively detailed this effect in chapter 4.3 of the book : *open-Space Microfluidics: Concepts, Implementations, Applications* by Delamarche and Kaigala (Wiley-VCH, 2018, doi:10.1002/9783527696789) as well as in the supplementary material of Safavieh et al, *Sci. Rep.* 2015.

Rather, the deformation is due to the fact that aspiration and injection distribution is not symmetric between the right and the left part of the device. For an element of fluid at the Q6/Q7 interface, the aspiration strength is stronger toward the right of the image. Hence the pattern deformation towards the right. The inconsistency comes from the theoretical image in Fig 1 d). While the streamline of this figure are the analytical solution, the background color defining the areas are handmade and were added later to highlight the different confinement areas. In the areas where the flow is low (close to stagnation points), the number of flow line is low and the interface between the two areas is imprecise. The interface between Q6/Q7 of Fig. 1d was approximated and should have been more curved. This is a difference between this figure and all others where the diffusion profiles are obtained mathematically to show the exact confinement areas.

Action taken : The background of the figure Fig. 1d was re-drawn. It has now been redone using a figure with more streamlines to give a more precise interface. A sentence was also added in the caption to precise that the red and green color background were added to highlight confinement areas. *“Theoretical streamlines (top) and fluorescence micrograph (bottom). Positive and negative sign respectively represent injection and aspiration apertures. To facilitate comparison between theory and experiment, green and red background were superposed on flow fields to highlight the expected areas of confinement not captured by streamlines”*

- SI – Fig 4 – Is it perhaps a similar effect that is seen on the ‘line’ pattern? (see previous)

[Authors] The effect is the same here too. The fluid velocity at each point can be seen as the sum of the velocity from every source (the dipoles are not isolated). Because it is a finite array of dipole, the aspiration strength is not symmetric. A dipole at the side of the array is affected by the other source on one side only. Because the aspiration are stronger than injection, the dipoles are curved toward the center. To have a perfectly straight line of dipoles you would need an infinite array, or to compensate the central aspiration by adding aspiration on the side of the dipole line. This problem and some potential solutions were explained in an article by Taylor, Zeaf, Lovchik and Kaigala. (Langmuir, 2016) (they use stabilisation aperture to “correct” the deformation, see Fig 4):

Taylor, D. P., Zeaf, I., Lovchik, R. D. & Kaigala, G. V. Centimeter-Scale Surface Interactions Using Hydrodynamic Flow Confinements. Langmuir 32, 10537–10544 (2016).

[Action taken] None.

- Methods - Exposure times for the fabrication.

[Authors] Details were added in the section Method - Microfluidic multipoles:

“MFMs were designed using Catia V5 (Dassault Systèmes, Vélizy-Villacoublay, France) and then 3D printed using a 27-micron resolution stereolithography printer (Freeform Pico and Pico 2 HD, Asiga, Alexandria, Australia). Plasgrey V2 resin (Asiga) and Pro3dure GR-1 resin (Pro3dure medical, Dortmund, Germany) were used to print the MFMs. Plasgrey V2 MFMs were printed using the Freeform Pico printer with a wavelength of 405 nm and an exposure time of 1.150 s. GR-1 MFMs were printed using the Pico 2 HD printer with a wavelength of 385 nm and an exposure time of 0.468s. A layer thickness of 50 µm were used for both resins.”

- Methods – Why were two materials used for making the devices?

[Authors] The “Pro3dure GR-1” resin allows fabrication of feature of smaller details than with the PlasGrey V2. The printing time is also slightly shorter. However, we started to work with this new resin (and new printer) after the majority of the experimental measurements were made. We used GR-1 resin only for the MFM in the immunoassay experiment.

We didn’t give so much details on the resin because we don’t consider it to be an important point of the article. Both resin (and many other) can be used to print MFMs. They are also commercially available.

[Action taken] None

- SI - .stl models of the MFM devices used in the work could be included in the SI

[Authors] We thank the reviewer for this excellent idea. We are happy to comply. The .stl will be added to the SI.

[Action taken] Sentence added at line 306 : “The MFMs .stl files are available in the supplementary material“

- Fig 3 (a-b) – this figure could benefit from being moved to an earlier point in the manuscript. In addition, simple labels for the CAD render and/or in the figure caption, would greatly enhance the figure.

[Authors] We understand the reviewers suggestion very well as we have debated extensively within our team as to when to introduce the device before submitting the paper. While it might improve the readability of the paper for an experimentally-oriented reader, we decided to opt for a “theory first” approach to emphasize this contribution. In the current situation, the figure is also placed in the paper (Fig. 3) “just in time” when we introduce the experimental setup in the text. We would honestly prefer to keep the figure at this place in the paper. However, if you believe that putting it earlier would greatly enhance the readability of the paper, we will comply and put it earlier.

[Action taken] Labels were added on Fig. 3a,b. Fig. 3b was changed to fit better with the labels.

- Fig 3 (c-h) – Considering that half of the experiment is covered in the figure at present, it is difficult to draw a full conclusion on the overlap of the MATLAB image. Perhaps it could be improved further by separating the theory, and experimental (as in Fig 1), but also including a full overlay of the two patterns?

[Authors] We have chosen to proceed with half figures in order to densify the amount of information, as there are many patterns to compare. No differences were observed between the top and bottom part of each images, so we believe that no information is lost using our representation. However, we are happy to comply with the reviewers suggestion by adding a figure with overlay in the SI (Supplementary Fig. 4). We also decided to add graph of fluorescence in function of position for differents lines on the images (Supplementary Fig. 3). These graphs allows to see the confinement areas as well as the diffusion length.

[Action taken] A figure with overlay comparison was added in the SI (Supplementary Fig. 4).

Supplementary Figure 4 Comparison between experimental figures and the analytical model. For each configuration, an experimental figure (left), an analytical model figure (middle left) an overlay between the experimental and analytical figure (middle right) and a side-by-side comparison (right) are shown. (a) Microfluidic dipole, (b) Polygonal multipole, (c) 8-petal axisymmetric “flower” multipole. Scale bars: 500 μm .

Graph of fluorescence in function of position were added in SI (Supplementary Fig. 3).

At present to the knowledge of the reviewer, the manuscript has only one reference to a 3D printed (3DP) microfluidic device. Considering that the conclusions result from direct comparison of theory with a 3DP microfluidic device, it would be pertinent to include a brief acknowledgement of advancement in that field, and how that 3DP is of benefit to the work described in the manuscript. There are numerous reviews on the subject (perhaps too many), but a number have been well received by the community.

[Authors]

We thank the reviewer for this useful suggestion. We have overlooked this important point as we had it covered in a previous article on 3D printing of MFs (Brimmo et al, *Sci. Rep.*, 2018). Yet, it is indeed a good idea to discuss the importance of additive manufacturing in the specific context of multipoles.

[Action taken] :

-We added a paragraph in the introduction to acknowledge the field and the recent works.

“On the experimental level, several MFM fabrication methods were presented over the years, but they all required microfabrication of silicon or glass components^{6,19,20} which are costly and slow to yield prototypes. In recent years, 3D printing has emerged as a disruptive technology for microfluidic device fabrication²¹⁻²³. 3D printing is fast and yields ready-to-use devices that require no alignment, bonding or molding steps. Moreover, single block printing of microfluidic devices affords full design flexibility in three dimensions that can simply not be realized using microfabrication processes dependent on iterative photolithography and microstructuring cycles. We recently presented a method for 3D printing of MFMs²⁴, but a proof of concept for the fabrication of more general open-space microfluidic devices is still missing.”

Cited references in previous section:

6 Juncker, D., Schmid, H. & Delamarche, E. Multipurpose microfluidic probe. *Nat. Mater.* **4**, 622–8 (2005).

- 19 Kaigala, G. V., Lovchik, R. D., Drechsler, U. & Delamarche, E. A Vertical Microfluidic Probe. *Langmuir* **27**, 5686–5693 (2011).
- 20 Lovchik, R. D., Drechsler, U. & Delamarche, E. Multilayered microfluidic probe heads. *J. Micromechanics Microengineering* **19**, 115006 (2009).
21. Waheed, S. *et al.* 3D printed microfluidic devices: enablers and barriers. *Lab Chip* **16**, 1993–2013 (2016).
22. Bhattacharjee, N., Urrios, A., Kang, S. & Folch, A. The upcoming 3D-printing revolution in microfluidics. *Lab Chip* **16**, 1720–1742 (2016).
23. Ho, C. M. B., Ng, S. H., Li, K. H. H. & Yoon, Y. J. 3D printed microfluidics for biological applications. *Lab Chip* **15**, 3627–3637 (2015).
24. Brimmo, A., Goyette, P., Alnemari, R., Gervais, T. & Qasaimeh, M. A. 3D Printed Microfluidic Probes. *Sci. Rep.* **8**, 10995 (2018).

-We modified a sentence (line 292) where we explained that we used 3D printing to fabricate MFM. The goal of this modification is to put emphasis why we used 3D printing and what were the advantages.

“To validate our theoretical model, multipolar microfluidic devices were fabricated using a previously published single step 3D rapid prototyping process²⁴, which offers several advantages over conventional silicon-based machining: it is fast and simple, and truly 3D structures can be produced, which allows the fabrication of compact MFM with high density of fluidic ports that are connected via complex fluidic routing architectures to arbitrarily positioned apertures.”

-We added a sentence in conclusion related to 3D printing of devices (line 475): “The limited resolution of commercial 3D printing technologies limits the integration of large numbers of connectors and multiplexers. However, custom 3D printer and resin⁴³ have been made demonstrating the integration of 88 connectors per mm², with a feature resolution < 20 μm^{44,45}, which could allow making MFM with higher resolution and higher density applications, and serve many more applications in the life sciences.”

43. Gong, H., Beauchamp, M., Perry, S., Woolley, A. T. & Nordin, G. P. Optical approach to resin formulation for 3D printed microfluidics. *RSC Adv.* **5**, 106621–106632 (2015).
44. Gong, H., Bickham, B. P., Woolley, A. T. & Nordin, G. P. Custom 3D printer and resin for 18 μm × 20 μm microfluidic flow channels. *Lab Chip* **17**, 2899–2909 (2017).
45. Gong, H., Woolley, A. T. & Nordin, G. P. 3D printed high density, reversible, chip-to-chip microfluidic interconnects. *Lab Chip* **18**, 639–647 (2018).

All the main points in the abstract, introduction, and conclusion are well outlined, except for the missing aspects mentioned previously about 3DP microfluidics.

The derivation of the mathematical equations for microfluidic multipoles is not within the reviewers current expertise; however, the mathematics themselves can be followed.

[Authors]: To improve the readability, shorten the text and respect *Nature Communications* format requirements, few minor changes were made throughout the manuscript. All changes are highlighted in yellow.

Reviewer #1 (Remarks to the Author):

The modifications made by the authors have helped clarify most of my original remarks.

However, I still have concerns about the conformal map from the physical plane to the potential (ϕ, ψ)-plane. Typically, transformations involving the logarithm give channel geometries (or, equivalently, ψ -periodic geometries) owing to the ' $2n\pi$ ' (n a integer) imaginary part of the logarithm. Thus I wonder if in fact the flow in the potential plane is past an periodic vertical stack of 'plates'? This does not affect the flow field, but it would affect the concentration field since the fundamental solution of Bazant (equation 6) applies only to a single plate along the (positive) real axis. Related to this, why do the exterior and interior problems map exclusively to the upper and lower half planes? This also implies that there is a discontinuity in c along the real axis which I am also uncomfortable with.

Given the above concerns then the argument given in the revised manuscript that $c=1/2$ on the dividing streamline is also doubtful. In any case the new argument presented by the authors leading to $c=1/2$ seems doubtful, since the square root map of the potential plane (giving the extensional flow) maps entirely to a half plane (only) and not the entire plane as the supplementary figure 1 suggests.

Hopefully the authors can clarify the above points.

Minor remark: In the revised manuscript, α is first used in line 176, but defined later (line 198).

Reviewer #2 (Remarks to the Author):

Many thanks to the authors for the time, and effort put into the updated manuscript. The quality of additions is most certainly to standard, and no further changes to the content are requested from this reviewer. It is appreciated that further inclusion of 3D printed microfluidic state-of-the-art is now discussed, both in the introduction and discussion. The inclusion of the new SI Figure 4 (which is also a remarkable image in its own right) illustrates the correlation between theoretical, and experimental results very clearly thank you. Also the addition of the .stl, and fabrication details is most welcome. They provide an extra level of clarity on the device design, particularly if the community attempts to recreate this work. It is also agreed that Fig 3(a-b) stay in the current order; the focus should stay on the theory initially.

Apologies that this comes a bit late, Christmas holidays and a new job have set things back a bit.

Reviewer #1 (Remarks to the Author):

The modifications made by the authors have helped clarify most of my original remarks.

However, I am still have concerns about the conformal map from the physical plane to the potential (ϕ, ψ)-plane. Typically, transformations involving the logarithm give channel geometries (or, equivalently, ψ -periodic geometries) owing to the ' $2n\pi i$ ' (n a integer) imaginary part of the logarithm. Thus I wonder if in fact the flow in the potential plane is past an periodic vertical stack of 'plates'? This does not affect the flow field, but it would affect the concentration field since the fundamental solution of Bazant (equation 6) applies only to a single plate along the (positive) real axis. Related to this, why do the exterior and interior problems map exclusively to the upper and lower half planes? This also implies that there is a discontinuity in c along the real axis which I am also uncomfortable with.

Given the above concerns then the argument given in the revised manuscript that $c=1/2$ on the dividing streamline is also doubtful. In any case the new argument presented by the authors leading to $c=1/2$ seems doubtful, since the square root map of the potential plane (giving the extensional flow) maps entirely to a half plane (only) and not the entire plane as the supplementary figure 1 suggests.

Hopefully the authors can clarify the above points.

Minor remark: In the revised manuscript, α is first used in line 176, but defined later (line 198).

We thank the reviewer for the thorough analysis of our results and the time they take giving pertinent and constructive comments.

As it stands, we identify 4 major concerns in the reviewer's response, which we will try to address here.

1 – Transformations involving logarithms typically give channel geometries, to which the “flow past a heated plate” solution cannot be applied directly.

We have solved the problem by assuming that the dipole geometry could be mapped to the entire plane in streamline coordinates, but indeed, after examining carefully the reviewer's argument, we agree now that this isn't the case. Parts of the real axis, which we take as an axis of symmetry, maps to the lines $\psi = \pi$ and $\psi = (1 - \alpha)\pi$. The streamline coordinate problem is thus not a semi-infinite absorber in the entire plane but rather a semi-infinite absorber confined in a channel with symmetry condition on both sides (ψ -periodic, as pointed out). This is something which we had not noticed and that forces us to revise one of our claims: that the model is exact for any Peclet number. Rather, we will argue here that it is asymptotically accurate for high enough Péclet numbers, a condition which is well respected by the usual parameters of operation of multipolar devices, in which the Péclet number usually ranges from 100 to 1000 or upwards for microfluidic devices transporting fluorescein or other equivalent solutes.

Furthermore, we will show below that the region in which the approximation breaks down is limited to a small distance from the aspiration aperture. In most practical situations, the region around the aspiration aperture is not the region of interest as the gradients there are extremely sharp compared to those around the stagnation point. Moreover, in all practical applications, the apertures are not truly

point-like but rather consist of circles or squares of finite size. In those cases, the region in which the approximation breaks down will be located under the finite aperture, a region that is already excluded from the model since we assumed point-like sources.

Calculating the break down point for a given Pe number

In streamline coordinates, the plate of constant concentration is confined between two axes of symmetry, one at $\psi = \pi$ and one at $\psi = (1 - \alpha)\pi$ (as stated in supplementary table 2), we will consider the one that is closest to the plate, which is, for $\alpha > 2$ at $\psi = \pi$. (The other case could easily be done using a similar argument and reach the same conclusion but in practice we always use $\alpha > 2$ as otherwise it is experimentally difficult to achieve correct confinement of the liquid). If the wake is far enough from the symmetry axis, we can assume that the unbounded space solution will be appropriate. We will consider that the wake is too close to the symmetry axis at the ϕ coordinate for which the concentration at $\psi = \pi$ predicted by the unbounded solution is 0.1 times the concentration of the plate. A similar development could be made for any other criterion (for example 0.01 times the concentration).

We thus have

$$\operatorname{erfc}\left(\operatorname{Im}\sqrt{\operatorname{Pe}(\Phi - \Phi_{stag})}\right) = 0.1$$

$$\operatorname{Im}\sqrt{(\Phi - \Phi_{stag})} \approx \frac{1.163}{\sqrt{\operatorname{Pe}}}$$

Taking $\omega = \sqrt{(\Phi - \Phi_{stag})}$, and labeling the real part of ω as ω_{real}

$$\omega = \omega_{real} + \frac{1.163}{\sqrt{\operatorname{Pe}}}i = \sqrt{\Phi - \Phi_{stag}}$$

$$\Phi - \Phi_{stag} = \left(\omega_{real} + \frac{1.163}{\sqrt{\operatorname{Pe}}}i\right)^2$$

$$\Phi - \Phi_{stag} = \left(\omega_{real}^2 - \frac{1.163^2}{Pe} \right) + 2\omega_{real} \frac{1.163}{\sqrt{Pe}} i = \phi + \psi i$$

By taking $\psi = \pi$, we can isolate ω_{real} and insert it in the expression for ϕ to find the position of the breakdown point. Thus, the unbounded solution is 0.1 times the plate concentration at $\psi = \pi$ for

$$\phi_{breakdown} - \phi_{stag} = \frac{Pe \pi^2}{4 \cdot 1.163^2} - \frac{1.163^2}{Pe}$$

For large enough Péclet numbers, we can consider only the term linear in Pe. We thus conclude that, in the streamline coordinate, the unbounded solution breaks down for $\phi - \phi_{stag} > \frac{\pi^2}{4 \cdot 1.163^2} Pe$, provided Pe is large.

Let's now see where this breakdown point maps in the (x, y) , dipole plane. Specifically, all values of ϕ over this breakdown value will map to the inside of a region around the aspiration aperture. We now want to see the size of this region.

We know that Φ is related to $z = x + y i$ by the complex potential, namely

$$\phi_{breakdown} - \phi_{stag} + \pi i = \log(z_b) - \alpha \log(z_b + 1)$$

For high Péclet numbers, we know that this will map to a region close to $z = -1$, and can thus consider only the term $-\alpha \log(z + 1)$, which will be exponentially bigger than the term $\log(z)$

$$x_b = \text{real}(z_b) = \exp\left(-\frac{1}{\alpha} (\phi_{breakdown} - \phi_{stag})\right) - 1$$

$$x_b + 1 = \exp\left(-\frac{\pi^2}{4 \cdot 1.163^2 \cdot \alpha} Pe\right)$$

We thus see that the distance between the breakdown point and the aspiration aperture (located at $x = -1$) decreases as $\exp\left(-\frac{\pi^2}{4 \cdot 1.163^2 \cdot \alpha} Pe\right)$. We plot this distance r_b vs Pe in the following figure

Figure 1 – Distance r_b of model breakdown from the aspiration aperture

In a dipolar device with $Pe = 10$, $\alpha = 4$, the “breakdown region” in which the model is no longer accurate is confined to a distance of 0.01 of the aspiration aperture. In practice, the aperture size is about a tenth of the distance between them, so the breakdown region is well under the apertures, a region in which the point-like sources and sink approximation already breaks down.

Thus, we revise our claim that the model is valid for any value of the Péclet number. In its place, we say that the model is valid for $Pe \gg 1$, which encompasses any practical situation in the domain of microfluidics. We leave to future work the determination of a more general model.

Actions taken: The text was modified (lines 202-212) so as to account for the presence of symmetry axes in streamline coordinates

In the streamline domain Φ , the problem is equivalent to a channel geometry with flows of concentration $c = 1$ and $c = 0$ separated by a no-flux boundary condition on the origin. At the stagnation point, the no-flux condition is dropped, and the flows are free to mix (Fig 2a) (see Supplementary Table 2 for more details on the streamline problem). The separating streamline going from the stagnation point to the aspiration aperture corresponds to the semi-infinite segment of the horizontal axis where the fluids can mix. If the Péclet number is high enough (higher than about 10, which is always realized in microfluidics applications), this segment can be taken to have concentration $c = 1/2$ and the walls of the channel geometry can be safely ignored (Supplementary Note 1). The

problem can thus be decomposed in two problems of advection-diffusion around semi-infinite obstacles of fixed concentration.

Moreover, any mention of “exact” model was removed or changed (line 24, line 107, line 129, line 429).

The precision “valid for high Péclet numbers” was added on line 239.

Also, the bulk of the argument given in this response has been put as supplementary note 1.

A sentence was added in the conclusion (line 423):

The analytical solutions provided are asymptotically accurate for high Péclet numbers ($Pe > 10$ in practice), which encompasses most applications of open-space microfluidics.

2 – It is unclear whether the streamline that passes through the stagnation point has constant concentration $C = \frac{1}{2}$.

We recognize that the argument that was given in the supplementary note was not precise enough. However, for high Péclet numbers, the separating streamline can be taken to have constant concentration $1/2$. Here is a justification:

In streamline coordinates, the problem is that of two flows of concentrations $c = 0$ and $c = 1$ separated by a no-flux boundary condition that disappears at the image of the stagnation point, so that the flows are free to mix on the semi-infinite segment of the horizontal axis $\phi > \phi_{stag}$. This semi-infinite segment corresponds to the separating streamline that goes through the stagnation point in the dipole flow. In the general problem, this segment will not necessarily have a constant concentration. However, if the flow is convective enough, we can neglect the influence of the symmetry axes at $\psi = \pi$ and $\psi = (1 - \alpha)\pi$, at least in a region $\phi < \phi_{breakdown}$. In that case, in the region where we neglect the influence of the symmetry axes, the problem is symmetrical around $\psi = 0$ and the concentration along the separating streamline is constant and equal to $1/2$, up to a certain exponentially small region around the aspiration aperture.

Action taken: Old supplementary note 1 has been removed, as it was unprecise. The specification that the separating line has concentration $1/2$ for highly convective regimes is given in the main text (line 208).

If the Péclet number is high enough (higher than about 10, which is always realized in microfluidics applications), this segment can be taken to have concentration $c = 1/2$ and the walls of the channel geometry can be safely ignored (Supplementary Note 1). The problem can thus be decomposed in two problems of advection-diffusion around semi-infinite obstacles of fixed concentration.

3 – Why do the inner and outer region map exclusively to the upper and lower half planes?

The fact that the inner and outer regions map to the lower and upper half-plane in streamline coordinates can be seen by checking that the stream function at the stagnation point is equal to 0. Since we separate the domain along the streamline, the image of the inner region in streamline coordinates will be in the lower half-plane while the outer region will be in the inner half-plane.

For the lower half-plane in dipole coordinates ($\text{Im}(z) < 0$), the opposite happens, with the inner region mapping to the upper half-plane and the outer region to the inner half-plane. This is due to the fact that $\text{Im}(\log(z^*)) = -\text{Im}(\log(z))$. For our purpose, we solve the problem for $\text{Im}(z) > 0$ and use symmetry to get the rest of the domain.

4- The presence of discontinuities in the streamline coordinate solution around the real axis is bothersome and appears unphysical

The presence of a discontinuity along the real axis in streamline coordinates for $\phi < \phi_{stag}$ is due to the fact that, when transformed to the dipole coordinates, both sides of this segment are “unzipped” and mapped to different segments in the dipole flow, one inside and one outside of the separating streamline. In streamline coordinates this would represent a condition of zero flux along the segment (which maps to a part of the symmetry axis in dipole coordinates) with a different value of the concentration at $\psi \rightarrow -\infty$ for the upper and lower part.

Another way of seeing it is that we are solving a different problem for the inner and outer region (since the upper and lower half-planes do not communicate in the streamline coordinate problem) and that we are combining both to obtain the full solution.

Minor remark: in the text, alpha is first used on line 176 but is defined later (line 198)

Action taken: “alpha > 1” changed for “aspiration superior to injection” on line 176. This way alpha is not used before it is introduced.

Reviewer #1 (Remarks to the Author):

I am happy with the authors' response and recommend publication.